# Nocturnal Hemodialysis Leads to Improvement in Physical Performance in Comparison with Conventional Hemodialysis

**DOI:** 10.3390/nu15010168

**Published:** 2022-12-29

**Authors:** Manouk Dam, Peter J. M. Weijs, Frans J. van Ittersum, Tiny Hoekstra, Caroline E. Douma, Brigit C. van Jaarsveld

**Affiliations:** 1Amsterdam UMC, Nutrition and Dietetics, Amsterdam Cardiovascular Sciences, VU University, De Boelelaan 1117, 1081 HV Amsterdam, The Netherlands; 2Nutrition and Dietetics, Faculty of Sports and Nutrition, Amsterdam University of Applied Sciences, Dr. Meurerlaan 8, 1067 SM Amsterdam, The Netherlands; 3Amsterdam UMC, Nephrology, Amsterdam Cardiovascular Sciences, VU University, De Boelelaan 1117, 1081 HV Amsterdam, The Netherlands; 4Spaarne Gasthuis Hoofddorp, Nephrology, Spaarnepoort 1, 2134 TM Hoofddorp, The Netherlands

**Keywords:** nocturnal hemodialysis (NHD), physical performance, protein-energy wasting (PEW), conventional hemodialysis (CHD), short physical performance battery (SPPB), quality of life (QOL), linear mixed models (LMM)

## Abstract

End-stage kidney disease patients treated with conventional hemodialysis (CHD) are known to have impaired physical performance and protein-energy wasting (PEW). Nocturnal hemodialysis (NHD) was shown to improve clinical outcomes, but the evidence is limited on physical performance and PEW. We investigate whether NHD improves physical performance and PEW. This prospective, multicenter, non-randomized cohort study compared patients who changed from CHD (2–4 times/week 3–5 h) to NHD (2–3 times/week 7–8 h), with patients who continued CHD. The primary outcome was physical performance at 3, 6 and 12 months, assessed with the short physical performance battery (SPPB). Secondary outcomes were a 6-minute walk test (6MWT), physical activity monitor, handgrip muscle strength, KDQOL-SF physical component score (PCS) and LAPAQ physical activity questionnaire. PEW was assessed with a dietary record, dual-energy X-ray absorptiometry, bioelectrical impedance spectroscopy and subjective global assessment (SGA). Linear mixed models were used to analyze the differences between groups. This study included 33 patients on CHD and 32 who converted to NHD (mean age 55 ± 15.3). No significant difference was found in the SPPB after 1-year of NHD compared to CHD (+0.24, [95% confidence interval −0.51 to 0.99], *p* = 0.53). Scores of 6MWT, PCS and SGA improved (+54.3 [95%CI 7.78 to 100.8], *p* = 0.02; +5.61 [−0.51 to 10.7], *p* = 0.03; +0.71 [0.36 to 1.05], *p <* 0.001; resp.) in NHD patients, no changes were found in other parameters. We conclude that NHD patients did not experience an improved SPPB score compared to CHD patients; they did obtain an improved walking distance and self-reported PCS as well as SGA after 1-year of NHD, which might be related to the younger age of these patients.

## 1. Introduction

Patients with end-stage kidney disease (ESKD) are in majority treated with conventional hemodialysis (CHD), regularly prescribed as 3–5 h of hemodialysis (HD) treatment thrice weekly, but are known to have poor health outcomes. Cardiovascular disease, hospitalization and mortality rates are high during CHD [1,2,3,4]. Additionally, physical performance levels are low in HD patients compared to healthy populations [5,6,7] and are found to be associated with poor quality of life (QOL), cardiovascular disease and mortality [8,9,10].

Therefore, twice as long HD sessions, often applied as nocturnal HD (NHD), were studied to improve outcomes [11] and appeared to have positive results on various clinical endpoints. For example, improvements are found in left ventricular hypertrophy, hypertension, hyperphosphatemia, use of erythropoiesis-stimulating agents, hospitalization rates and survival, when comparing NHD to CHD [12,13,14,15,16,17,18,19,20,21]. However, limited studies have investigated the effect of NHD on physical performance and showed contradictory results [22]. The frequent hemodialysis network (FHN) trial reported no improvement in physical performance in NHD patients [23]. However, prospective cohort studies found significant improvements in exercise duration and walking distance [24,25], but self-reported physical performance showed both positive changes as well as neutral effects [14,21,23,25,26]. Therefore, it remains unclear whether an NHD regimen, compared to CHD, can improve physical performance.

When investigating physical performance, factors regarding protein-energy wasting (PEW) should be taken into account as well. PEW contributes to poor health outcomes [27,28] and is characterized by multiple factors, such as decreased levels of physical activity, protein intake and muscle mass [27,28,29]. Improvement in physical performance would be difficult to attain when patients have poor protein intake and PEW. Some parameters of PEW have been investigated in NHD patients and most studies seem to show improvements in body weight and protein intake compared to CHD [17,30,31,32,33,34]. However, little is known regarding other aspects of PEW, such as muscle mass or appetite.

Therefore, the primary aim of this study is to investigate whether physical performance improves in patients who change from CHD to NHD, compared to patients who continue CHD treatment. Secondarily, we investigate if aspects of PEW improve during NHD treatment.

## 2. Materials and Methods

### 2.1. Study Design

This study is a prospective, multicenter, observational study. Patients were recruited from DiaPriva Dialysis Center, Amsterdam UMC (location VU University Medical Center) and Spaarne Hospital, the Netherlands, between June 2014 and 2019. Details regarding this study were published previously [35]. The study protocol is in accordance with the Declaration of Helsinki and has been approved by the Medical Ethics Committee of Amsterdam UMC (https://www.trialregister.nl/trial/4490, accessed on 12 July 2021).

### 2.2. Dialysis Treatments

Inclusion criteria were: (i) ≥18 years old, (ii) stable HD treatment of ≥3 months (iii) informed consent and (iv) ability to understand the study protocol. Exclusion criteria were: (i) dementia, (ii) life expectancy of fewer than 12 months, (iii) a planned renal transplantation within 12 months, (iv) unstable angina pectoris, (v) recent myocardial infarction, (vi) severe pulmonary disease and (vii) treatment incompliance, i.e., non-adherence to dialysis regimen. Patients who preferred to switch from CHD to NHD were assessed by their nephrologist to evaluate whether NHD was medically feasible. If so, they were not randomized, but considered eligible for the NHD group (2–3 times/week, 7–8 h per session) and asked if they wanted to participate in this study. Patients who did not want to be treated with NHD for personal reasons (e.g., because of social reasons or personal preferences) but met the inclusion criteria, were considered eligible for the control group. In order to avoid inclusion bias, only patients who were medically stable (i.e., without unstable angina pectoris, recent acute coronary syndrome, recent epileptic insult, etc.) were included in the study. The CHD patients continued a 2–4 weekly schedule, 3–5 h per session.

### 2.3. Outcomes

Assessments were repeated at baseline and in order to prevent a possible learning effect of the physical performance tests, the second of two assessments was used for analysis. After baseline measurements, the NHD group switched from CHD to NHD. After 3, 6 and 12 months measurements were repeated. The measurements were performed during or after HD, depending on the character of the measurement.

#### 2.3.1. Physical Performance

The primary outcome was physical performance, measured with the Short Physical Performance Battery (SPPB) [36] consisting of (i)two gait speed tests using a 4-minute walk with the fastest value used for analyses, (ii)five repeated chair stands and (iii)three balance tests of 10 s each (side-by-side, semi-tandem and tandem position). Higher scores represent better physical performance (0–12 points).

Secondary outcomes of physical performance were: the 6-minute walk test (6MWT), a 7-day physical activity monitor (PAM), handgrip muscle strength (HGS), a Kidney Disease Quality of Life-Short Form questionnaire (KDQOL-SF) physical component score (PCS) and a LASA physical activity questionnaire (LAPAQ). The 6MWT was performed indoors, on a walking course of 30 m and measured the total distance walked by the patient in 6 minutes [37]. The PAM (PAM B.V., Oosterbeek, The Netherlands) measures different activity levels during the day, including exercise activities, and provides a score of activity in minutes/day. HGS was assessed twice with a hand dynamometer (JAMAR^®^, Chicago, IL, USA) and the highest value was used for analysis. PCS (0–100) was derived from the KDQOL-SF (Dutch version 1.2). The LAPAQ [38] contained questions regarding walking, cycling, gardening, sports (including physical rehabilitation) and household activities over the last two weeks and total time spent on activities (minutes/day) was used for analyses.

#### 2.3.2. Protein-Energy Wasting

The secondary outcome PEW was assessed with: a dietary record, body weight, body mass index (BMI), dual-energy X-ray absorptiometry, bioelectrical impedance spectroscopy (BIS) and subjective global assessment (SGA), appetite, mid-upper arm muscle circumference (MUAMC) and serum albumin.

From the 3-day dietary record, mean protein, phosphorus and energy intake were derived at baseline and after 1 year. Dry body weight was established by the nephrologist at every assessment. Dual-energy X-ray Absorptiometry (DXA, Discovery A (S/N84993), Hologic Netherlands B.V., Amsterdam, The Netherlands) was performed at baseline and after 1-year. Information regarding fat mass in kg (FM) and appendicular skeletal muscle mass (ASMM) in kg (sum of lean mass in arms and legs without bone and fat mass) was used. FM and lean tissue mass (LTM) were measured with BIS (Body Composition Monitor, Fresenius Medical Care, Bad Homburg, Germany). Measurements were performed at least 30 minutes after a patient’s treatment, to create optimal stability in a patient’s body fluid compartments. Appetite was assessed by a visual analogue scale (VAS, 0–10). Mid-upper arm circumference (MUAC) was measured and triceps skinfold (TSF) using a skinfold caliper (Harpenden, Baty International, West Sussex, United Kingdom). MUAMC was calculated by the equation: MUAC (cm) − (3.14 × TSF (cm)). A 7-point SGA was used, with higher scores (6–7) indicating a normal nutritional status.

An overall PEW score was determined at baseline and after 1-year of treatment, by the presence of at least three of the following four criteria(28): (i)serum albumin <3.8 g/100 mL, (ii)BMI <23 kg/m^2^, (iii)reduced MUAMC (>10% compared with the 50th percentile of the reference population) and (iv)protein intake <0.8 g/kg/day. The criteria of reduced muscle mass over 3 or 6 months was excluded since repeated measurements could cross over. Albumin was measured before dialysis with either the bromcresol green or purple method (bromcresol purple values were converted) [39].

#### 2.3.3. Quality of Life and Sleep Parameters

QOL was assessed with the KDQOL-SF, containing a generic part (SF-36) and a disease-specific part, which lead to a PCS and mental component score (MCS), ranging from 0 to 100. Higher scores represent better QOL [40]. Additionally, the questionnaire provides 8 multi-item scales that were included in the analyses [40]. At last, a sleep questionnaire was obtained on sleep quality (score 0–9), causes of sleep disturbance (score 0–6) and effects of sleeping disturbance on day-time functioning (score 0–9) and two VAS (0–10) were filled out regarding restless legs/anxiety and tiredness after HD. Lower scores represented better sleeping behaviors.

### 2.4. Statistical Analysis

#### 2.4.1. Sample Size

A clinically significant difference of 10% in physical performance measured with the SPPB was estimated to be a relevant finding. With a statistical significance level of 0.05, a power of 80% and a standard deviation (SD) of 1.2–1.5 of the SPPB, a sample size of 16–25 patients per group were calculated to be required [41]. The power of the study is increased due to four repeated measurements within a patient, which allows for some attrition. Therefore, the aim was to include at least 50 patients.

#### 2.4.2. Analysis of Outcomes

Normally distributed variables were reported as mean ± SD, skewed variables as median with interquartile ranges and categorical data as number (percentage). Comparisons of normally distributed variables were performed with independent student *t*-tests, skewed variables with Mann–Whitney U tests and categorical data with chi-square tests.

All outcomes were analyzed according to an intention-to-treat strategy, with linear mixed models (LMM). Adjustments were made for the dependency of repeated observations within the individual, by adding a random intercept to the model. Treatment, time and interaction of treatment*time were defined as fixed variables because we hypothesized that there would be a difference in outcome between treatment groups and over time. For most outcomes, we assumed that development over time would be linear, but we discussed that some outcomes might be influenced mostly during the first 6 months of NHD, after which a plateau of the effect would be reached (e.g., VAS appetite or 6-MWT). In contrast, for some outcomes effects might occur only after 6 months of NHD (e.g., body composition). Therefore, we analyzed time in both ways: time as a continuous variable and time represented as dummy variables, also in order to detect possible non-linear developments over time.

Gender, age, history of renal replacement therapy (RRT vintage, defined as the total period of renal replacement therapy in the form of dialysis treatment and/or renal transplantation) and diastolic blood pressure were considered to affect outcomes and were therefore analyzed as covariates. Diastolic blood pressure was added into the models of physical performance and QOL because we expected this variable to possibly influence only the relations of those outcomes. All variables, except for diastolic blood pressure, were identified as confounders and added to all adjusted models. Prior to analyzing confounders, we investigated gender as a possible effect modifier. Random slopes were not added to the model, because it led to non-converging models. Estimated marginal means (EMM) were calculated, because this provides more insight into the results, instead of the presentation of mere regression coefficients. EMM include the random effect and covariates within the model. Data of the total group are presented, as results were comparable between males and females. The LMM analysis is presented with time as a continuous variable because it appeared to show the same results as models with time as dummy variables. Statistical significance was defined as *p* ≤ 0.05 and performed with SPSS (version 26.0, Chicago, IL, USA).

## 3. Results

### 3.1. Recruitment

In total, 127 patients were assessed for eligibility, of whom 65 were included. Of those, 32 patients switched from CHD to NHD and 33 patients remained on CHD treatment (Figure 1). Due to renal transplantation (*n* = 10), discontinuation of NHD (*n* = 8), loss to follow-up (*n* = 3) and death (*n* = 2), 3-month data were available for 58 patients, 6-month data for 51 patients and 1-year data for 42 patients. During the study, 47 serious adverse events were reported by the investigator (all hospitalizations), but determined by the steering committee as not related to study procedures.

### 3.2. Baseline Characteristics and Dialysis Treatments

The CHD group had a mean age of 60 ± 13.9 years and 49% were male, and the NHD group had a mean age of 50 ± 15.3 years and 56% males. Age and diastolic blood pressure were statistically significantly different between groups at baseline (*p* = 0.01 and *p* = 0.02, respectively). No other significant differences were found (Table 1). In Table 2 differences in dialysis treatments are presented.

### 3.3. Physical Performance: Differences between NHD and CHD

In Table 3 results on physical performance are presented for NHD patients in comparison with CHD patients. The SPPB showed no significant effect after 1 year of NHD treatment (+0.24, 95% confidence interval (CI) −0.51 to 0.99, *p* = 0.7), in comparison with CHD. Walking distance (6MWT) showed an overall improvement of 54.3 m (95% CI 7.78 to 100.8, *p* = 0.04) after 1 year of NHD, compared to CHD, also presented in Figure 2A. PCS increased by 5 points in NHD patients, whilst the CHD group decreased by 1 point (+5.61, 95% CI −0.51 to 10.7, *p* = 0.03).

### 3.4. Protein-Energy Wasting: Differences between NHD and CHD

In Table 4 results are presented for NHD in comparison with CHD patients on aspects of PEW. SGA showed a significant effect after 1-year in NHD patients (+0.71, 95% CI 0.36 to 1.05, *p* <0.001). ASMM appeared to decrease less in NHD compared to CHD patients (−0.5 kg in NHD versus −1.1 kg in CHD), although not significantly. Similar results were found for LTM, as the NHD group decreased by −0.4 kg after 1 year and CHD by −2.1 kg. Appetite score increased by 0.7 points after 1 year of NHD, in comparison with 0.1 points in CHD patients. Albumin improved non-significantly with 0.5 points during NHD, whilst decreased with −0.1 points in CHD. C-reactive protein (CRP) showed slightly lower means in the NHD group, but especially after 6 months a large decrease of 36% was found in NHD patients and a large increase of 33% in the CHD group. No significant effect was found in CRP after 1-year in NHD patients compared to CHD (−1.66, 95% CI −8.35 to 5.03, *p* = 0.6). Almost no patients with overall PEW were found (NHD; *n* = 1 at baseline and *n* = 0 after 1 year, CHD; *n* = 0 at baseline and *n* = 2 after 1 year).

### 3.5. QOL and Sleep: Differences between NHD and CHD

Table 5 shows the differences between NHD and CHD of QOL and sleep parameters. MCS was unchanged after 1-year of NHD (−1.65, 95% CI −8.23 to 4.93, *p* = 0.6). The overall health score improved by 7.2 points in 1-year, compared to a reduction of 2 points in CHD patients (Figure 2C). The KDQOL-SF multi-item scales showed similar results between groups, except for the burden of kidney disease which improved by 11.9 points (95% CI 0.34 to 23.6, *p* = 0.04, Figure 2D). The sleep parameters showed no differences between groups.

## 4. Discussion

In this study, we investigated whether patients who switched from CHD to NHD treatment improved their physical performance and PEW compared to patients continuing with CHD. Our primary outcome, physical performance measured by the SPPB, showed no change after 1-year of NHD. We did see differences in other physical performance tests: walking distance and self-reported physical performance improved in NHD compared to CHD patients, while no difference was found in daily physical activity and strength. Additionally, concerning PEW an improvement in SGA was found. Dietary intake, body composition and appetite showed no significant improvement by NHD treatment. Changes in ASMM and LTM as well as appetite indicated changes in line with SGA.

The limited number of studies have investigated the effect of NHD on aspects of physical performance and found different results. We found no change in the SPPB score and this was in accordance with the study by Hall et al. This FHN randomized controlled nocturnal trial lacked to find improvement in physical performance measured by the SPPB after 1-year. Hall et al. described that their power was probably too low to detect small changes or to detect changes in subgroup analyses [23]. The SPPB primarily assesses lower extremity function with three moderate effort tests and seems well suitable to detect changes in the elderly population [36,42]. In the FHN trial, subjects were younger (mean age 52–54 years old) and started with relatively high SPPB scores at baseline (approx. 8–9) and therefore, only modest changes could be detected during follow-up. Similar results were seen in our study, with a mean age of 55 years and SPPB scores of approx. 9.5 at the start. Therefore, with high scores already indicating a relatively good performance at baseline, any further improvements in SPPB scores could not be found, possibly because SPPB was not sensitive enough in our patients for small changes and a ceiling effect occurred [43].

We also assessed physical performance with a 6MWT and found improvement of walking distance in NHD patients after 1-year, whilst the CHD group remained stable. Two previous observational, prospective cohort studies investigated walking distance, exercise duration and capacity in NHD patients and found an improvement after 6 to 12 months [24,25]. The 6MWT is a more physically intensive test, as compared with the SPPB. Therefore, we think this assessment turned out to be more responsive for changes in our patient group.

In addition to the 6MWT, PCS significantly improved after 1-year. Additionally, the domain burden of kidney disease also improved. A few studies investigated physical performance with self-reported measurements during NHD, but results were inconsistent [14,23,25,44]. The variability among these studies could be caused by several factors, such as study design, NHD in a home setting or frequent NHD (6 times/week). Patients treated with NHD 2 to 3 times/week, who sleep adequately in separate bedrooms at our facility, indeed reported improved energy, and more spare time enabling participation to social activities. We conclude that this schedule of 2 to 3 times/week, has positively influenced patients’ perspective on the burden of their disease and PCS.

We found a significant improvement in SGA score and to our knowledge, no further research investigated SGA in NHD patients. SGA is a validated tool to assess nutritional status [45], including aspects of appetite, weight loss, dietary intake and muscle wasting. This improvement is an important effect of NHD treatment. Lower SGA scores represent worse nutritional status and were found to be associated with an increased risk of mortality [46]. Although our results on ASMM, LTM, appetite and albumin were not statistically significant, the measurements show results of the same order. ASMM and LTM decreased in both groups, but far less in the NHD group compared to CHD. Appetite scores and albumin improved, whilst in the CHD group remained stable. Since 43 patients were left in our 1-year analyses, the power was probably too low to demonstrate significant changes in all assessments of PEW. When analyzing an overall PEW, we found almost no patients with PEW, which is not in accordance with previous numbers, since PEW is a well-known problem in HD patients [29]. This could be the result of a somewhat younger patient group, and possibly the fact that we did not include time-depending criteria.

No difference was found in HGS, body weight, dietary intake or an increase in muscle mass, which seem to be in accordance with other studies [17,30,31,34,47,48]. Perhaps, it might be difficult to expect changes in strength or body composition, without offering exercise and nutritional intervention. Previous studies on HD patients showed improvement in strength and muscle mass when patients followed an exercise program [49]. No difference was found in daily activities measured with PAM or LAPAQ, although our data did show that 6 patients in the NHD group started with exercise activities in comparison with 1 CHD patient during our study. We found that NHD patients in our study performed better, i.e., by showing an improved walking distance and felt better by showing an improved PCS and SGA. The fact that more NHD patients even started with exercise or rehabilitation activities, might be the result of the combination of having more physical energy and more spare time during the day due to the NHD. On the other hand, the NHD patients did not actually improve their daily activities, as reported by the PAM and LAPAQ, which we cannot fully explain.

This study has several strengths. To our knowledge, this is the first study investigating physical performance and PEW in NHD patients, using a broad range of tests. Secondly, the performance of two baseline measurements, in order to prevent a learning effect of the physical tests, adds to the reliability of our results. Third, data analysis by LMM is the optimal way to investigate the outcome of interest.

This study also had limitations. Patients were not randomized to CHD or NHD, which resulted in a lower age for the NHD group at baseline. Additionally, we did not correct for multiple testing, however, with a small sample size of 43 patients left after 1 year of NHD, it appears unlikely that this could explain the improvement in SGA. Additionally, dialysis vintage was found to be non-significantly lower in the NHD group at baseline. In general, age and dialysis vintage would be expected to influence elements of physical performance and PEW, so therefore all analyses were adjusted for age, dialysis vintage and gender. Additionally, patients who did not want to be treated with NHD because of personal reasons were considered for the control group, as we wanted to avoid including patients who were excluded from NHD due to medical reasons. Personal reasons often had to do with patients not wanting to sleep elsewhere. Even so, perhaps it partly reflects a patient’s motivation regarding kidney replacement therapy and could therefore have led to a different population in the CHD group.

## 5. Conclusions

In conclusion, treatment of NHD compared to CHD resulted in a comparable SPPB score; we did find improvement in 6-minute walking distance, self-reported PCS and SGA scores in patients treated with 1 year of NHD; however, these might be related to the younger age of patients treated with this modality.

## Figures and Tables

**Figure 1 nutrients-15-00168-f001:**
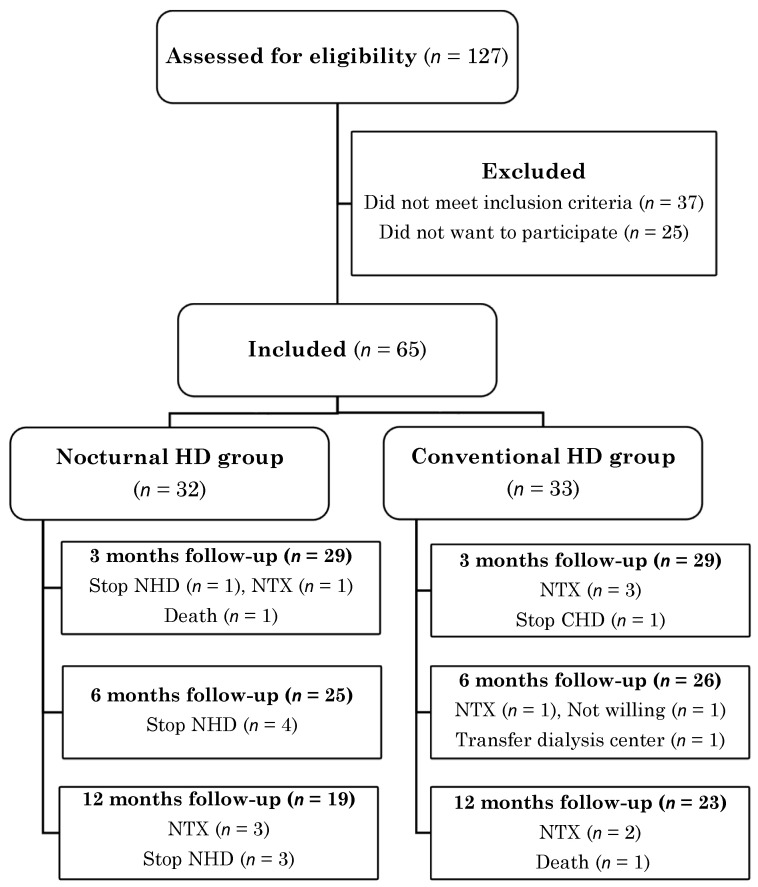
Participant flow diagram.

**Figure 2 nutrients-15-00168-f002:**
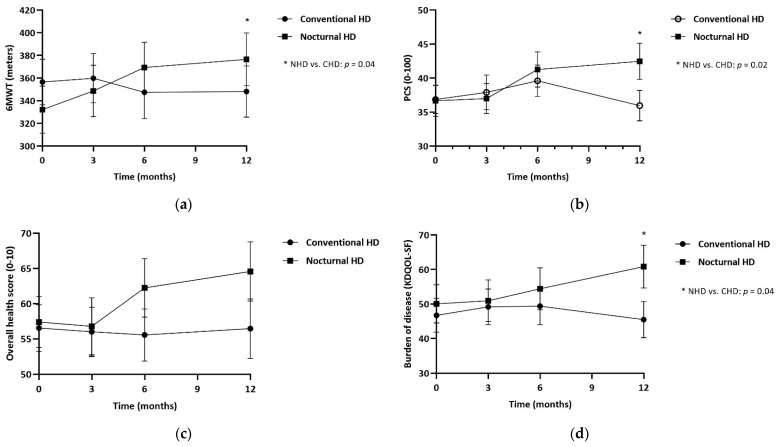
Estimated marginal means at 3, 6 and 12 months, for patients treated with NHD and CHD. (**a**) 6-minute walk test in meters; (**b**) Physical component summary score (PSC score, 0–100) by KDQOL-SF; (**c**) Overall health score (0–10) by KDQOL-SF; (**d**) Burden of kidney disease by KDQOL-SF. * *p* ≤ 0.05.

**Table 1 nutrients-15-00168-t001:** Patient characteristics at baseline.

Characteristics	CHD (*n* = 33)	NHD (*n* = 32)	*p*-Value
Male, n (%)	16 (49%)	18 (56%)	0.4
Age, years	60 ± 13.9	50 ± 15.3	0.01
Race, n (%) White Black Asian Other/mixed	21 (63.6%) 3 (9.1%) 3 (9.1%) 6 (18.2%)	19 (59%) 8 (25%) 1 (3%) 4 (12.5%)	0.3
Causes of End-Stage Kidney Disease, n (%) Diabetes mellitus Hypertension Polycystic kidney disease Other	8 (24%) 6 (18%) 1 (3%) 18 (55%)	5 (16%) 6 (19%) 3 (9%) 18 (56%)	0.6
Charlson comorbidity index, n (%) 2 (no comorbidity except ESKD) 3–4 (intermediate comorbidity) ≥5 (high comorbidity)	7 (21.2%) 14 (42.5%) 12 (36.4%)	11 (34.4%) 11 (34.4%) 10 (31.3%)	0.5
Body weight, kg	77.4 ± 18.6	81.8 ± 20.3	0.4
Body Mass Index, kg/m^2^	27 ± 6.4	28 ± 6.7	0.5
Systolic blood pressure, mm Hg	134 ± 24.7	141 ± 21.9	0.3
Diastolic blood pressure, mm Hg	72 ± 12.4	79 ± 12.3	0.02
History of renal replacement therapy (RRT vintage), months	40 (15–123)	21 (9–75)	0.1
Previous renal transplant, n (%)	10 (30.3%)	9 (28.1%)	0.9
Anuria, n (%)	16 (48.5%)	10 (31.3%)	0.3

Data presented as mean ± SD, as median with IQR or as percentages when appropriate. Independent *t*-tests were used for normally distributed variables, Mann–Whitney U tests for skewed data and chi-squared tests for categorical variables.

**Table 2 nutrients-15-00168-t002:** Characteristics of dialysis treatments.

Characteristics	CHD (*n* = 33)	NHD (*n* = 32)	*p*-Value
Dialysis frequency, times/week			
Baseline	2.8 ± 0.5	2.9 ± 0.3	0.3
6 months	2.9 ± 0.5	2.9 ± 0.3	1
12 months	2.9 ± 0.4	2.8 ± 0.4	0.8
Dialysis duration per session, hours			
Baseline	3.9 ± 0.4	3.9 ± 0.3	0.5
6 months	3.9 ± 0.4	8.0 ± 0.2	<0.001
12 months	3.9 ± 0.5	7.9 ± 0.3	<0.001
Dialysis dose (standardized Kt/V week)			
Baseline	5.9 ± 1.3	6.0 ± 1.0	0.8
6 months	6.5 ± 1.1	7.6 ± 0.9	<0.001
12 months	6.8 ± 1.2	7.9 ± 0.6	<0.001

Data presented as mean ± SD, analyzed with independent *t*-tests.

**Table 3 nutrients-15-00168-t003:** Physical performance over 1 year in nocturnal hemodialysis patients compared to conventional hemodialysis patients.

Physical Performance						
	Baseline	3 Months	6 Months	12 Months	Linear Mixed Models with 1-Year Treatment Effect of NHD Compared to CHD *	
	Mean (SE) ^a^	Mean (SE) ^a^	Mean (SE) ^a^	Mean (SE) ^a^	Effect (95% CI)	*p*-Value
Short physical performance battery, scale, 1–12 CHD NHD	9.53 (0.29) 9.59 (0.31)	9.42 (0.31) 9.78 (0.34)	9.42 (0.32) 9.57 (0.34)	9.66 (0.33) 10.1 (0.35)	0.24 (−0.51 to 0.99)	0.5
6-minute walk test, meters CHD NHD	356 (20.3) 332 (20.9)	360 (21.8) 349 (22.8)	347 (23.3) 369 (22.4)	348 (22.7) 377 (23.4)	54.3 (7.78 to 100.8)	0.02
Physical activity monitor, min/day CHD NHD	63.2 (8.28) 78.7 (8.53)	51.2 (10.1) 81.9 (10.9)	37.7 (11.7) 76.6 (10.6)	47.1 (10.3) 71.4 (11.2)	14.7 (−14.6 to 44.2)	0.3
Handgrip muscle strength, kg CHD NHD	30 (1.38) 29 (1.47)	30 (1.41) 30 (1.54)	31 (1.44) 29 (1.54)	30 (1.44) 30 (1.57)	1.35 (−0.98 to 3.68)	0.3
Physical component score, score 0–100 CHD NHD	37 (2.07) 37 (2.31)	38 (2.55) 37 (2.20)	40 (2.31) 41 (2.58)	36 (2.24) 42 (2.65)	5.61 (−0.51 to 10.7)	0.03
LAPAQ activity questionnaire, min/day CHD NHD	122 (15.4) 123 (16.2)	115 (15.9) 138 (18.1)	103 (17.5) 117 (18.2)	102 (17.2) 112 (19.5)	12.6 (−33.9 to 59.2)	0.6

^a^ The observed data are reported as estimated marginal means (SE) values, including the random effect and adjustment for covariates within the model. * Differences (95% CI) and *p*-values are presented for the comparison between NHD and CHD, calculated with linear mixed model analysis of repeated measures. The overall 1-year effect is presented for the treatment interaction over time. Fixed factors included time, treatment and time * treatment interaction. Random intercepts included subject. In all models covariates gender, age and history of renal replacement therapy (RRT vintage) were added.

**Table 4 nutrients-15-00168-t004:** Aspects of protein-energy wasting over 1-year in nocturnal hemodialysis patients compared to conventional hemodialysis patients.

Protein-Energy Wasting						
	Baseline	3 Months	6 Months	12 Months	Linear Mixed Models with 1-Year Treatment Effect between NHD and CHD *	
	Mean (SE) ^a^	Mean (SE) ^a^	Mean (SE) ^a^	Mean (SE) ^a^	Effect (95% CI)	*p*-Value
Protein intake, g/day CHD NHD	70.4 (3.94) 76.4 (4.39)	-	-	69.7 (4.51) 76.0 (4.99)	0.33 (−12.9 to 13.6)	1.0
Protein, g/kg CHD NHD	0.94 (0.07) 1.00 (0.08)			0.98 (0.08) 1.02 (0.08)	−0.08 (−0.03 to 0.10)	0.4
Phosphorus intake, mg/day CHD NHD	1159 (102) 1367 (113)	-	-	1267 (123) 1334 (135)	−142 (−577 to 292)	0.5
Energy intake, kcal/day CHD NHD	1758 (106) 1870 (117)	-	-	1637 (136) 1916 (123)	167 (−222 to 557)	0.4
Body weight, kg CHD NHD	77.1 (3.30) 82.0 (3.53)	77.1 (3.31) 80.8 (3.55)	76.8 (3.32) 80.5 (3.55)	76.4 (3.31) 79.9 (3.56)	−1.39 (−3.14 to 0.35)	0.1
Body mass index, kg/m^2^ CHD NHD	26.7 (1.11) 28.4 (1.18)	26.7 (1.11) 28.0 (1.19)	26.6 (1.11) 27.9 (1.19)	26.5 (1.19) 27.7 (1.12)	−0.46 (−1.08 to 0.13)	0.1
DXA- Appendicular skeletal muscle mass, kg CHD NHD	21.2 (1.05) 22.3 (0.95)	-	-	20.1 (1.08) 21.8 (0.99)	0.53 (−0.64 to 1.71)	0.3
DXA- Fat mass, % CHD NHD	34.4 (1.40) 34.4 (1.35)	-	-	35.8 (1.51) 35.4 (1.41)	−0.37 (−2.49 to 1.74)	0.7
BIS- Lean tissue mass, kg CHD NHD	39.3 (1.40) 37.3 (1.40)	38.3 (1.45) 35.4 (1.51)	38.8 (1.53) 36.9 (1.51)	37.2 (1.57) 36.9 (1.60)	1.29 (−1.98 to 4.58)	0.4
Appetite, VAS 0–10 cm CHD NHD	7.5 (0.37) 7.3 (0.39)	7.3 (0.38) 7.5 (0.44)	7.1 (0.41) 7.7 (0.44)	7.6 (0.42) 8.0 (0.47)	0.78 (−0.33 to 1.90)	0.2
Subjective global assessment, score 1–7 CHD NHD	5.8 (0.13) 5.6 (0.14)	5.7 (0.13) 6.1 (0.15)	5.7 (0.15) 6.2 (0.14)	5.9 (0.16) 6.3 (0.14)	0.71 (0.36 to 1.05)	<0.001
Mid-upper arm muscle circumference, cm CHD NHD	25.4 (0.59) 25.1 (0.63)	25.4 (0.59) 25.8 (0.66)	25.3 (0.62) 25.4 (0.66)	25.1 (0.62) 25.1 (0.67)	0.46 (−0.59 to 1.52)	0.4
Albumin, g/L CHD NHD	40.3 (0.52) 40.2 (0.56)	40.4 (0.59) 39.9 (0.59)	39.8 (0.59) 41.3 (0.54)	40.2 (0.55) 40.7 (0.61)	1.04 (−0.07 to 2.14)	0.06
C-reactive protein, mg/L CHD NHD	8.74 (2.03) 7.77 (2.32)	9.03 (2.13) 7.62 (2.47)	11.6 (2.27) 4.99 (2.44)	7.63 (2.32) 7.53 (2.73)	−1.66 (−8.35 to 5.03)	0.6

^a^ The observed data are reported as estimated marginal means (SE) values, including the random effect and covariates within the model. * Differences (95% CI) and *p*-values are presented for the comparison between NHD and CHD, calculated with linear mixed model analysis of repeated measures. The overall 1-year effect is presented for the treatment interaction over time. Fixed factors included time, treatment and time * treatment interaction. Random intercepts included subject. In all models covariates gender, age and history of renal replacement therapy (RRT vintage) were added.

**Table 5 nutrients-15-00168-t005:** Quality of life over 1 year in nocturnal hemodialysis patients and conventional hemodialysis patients.

Quality of Life						
	Baseline	3 Months	6 Months	12 Months	Linear Mixed Models with 1-Year Treatment Effect between NHD and CHD *	
	Mean (SE) ^a^	Mean (SE) ^a^	Mean (SE) ^a^	Mean (SE) ^a^	Effect (95% CI)	*p*-Value
Mental component summary score, scale 0–100 CHD NHD	53 (1.96) 53 (2.16)	54 (2.18) 51 (2.57)	51 (2.37) 50 (2.62)	51 (2.25) 50 (2.73)	−1.65 (−8.23 to 4.93)	0.6
Overall health score, scale 0–10 CHD NHD	56.5 (3.30) 57.4 (3.65)	56.0 (3.50) 56.8 (4.05)	55.6 (3.68) 62.3 (4.15)	56.5 (3.60) 64.6 (4.21)	7.10 (−1.08 to 15.3)	0.08
Sleep quality, score 0–9 CHD NHD	2.22 (0.32) 2.09 (0.33)	1.52 (0.33) 2.15 (0.37)	1.54 (0.54) 2.04 (0.37)	1.66 (0.35) 1.33 (0.41)	0.15 (−0.82 to 1.11)	0.8
Causes of sleep disturbances, score 0–6 CHD NHD	0.63 (0.19) 0.85 (0.20)	0.96 (0.19) 0.90 (0.23)	0.88 (0.22) 0.71 (0.23)	0.72 (0.22) 0.56 (0.53)	−0.41 (−1.05 to 0.23)	0.2
Effect of sleeping disturbance on day-time functioning, score 0–9 CHD NHD	2.26 (0.35) 2.79 (0.37)	2.38 (0.43) 3.92 (0.36)	2.72 (0.39) 3.02 (0.42)	2.95 (0.39) 3.11 (0.45)	−0.32 (−1.39 to 0.76)	0.6
Restless legs/anxiety during sleep, VAS 0–10 cm CHD NHD	2.62 (0.45) 2.07 (0.47)	2.16 (0.47) 2.05 (0.55)	1.49 (0.52) 2.36 (0.55)	1.52 (0.52) 1.85 (0.62)	1.19 (−0.43 to 2.83)	0.2
Fatigue after sleep, VAS 0–10 cm CHD NHD	4.47 (0.78) 5.08 (0.82)	4.66 (0.95) 5.03 (0.82)	6.05 (0.91) 4.77 (0.95)	5.00 (1.06) 3.63 (0.91)	−2.03 (−4.88 to 0.83)	0.2

^a^ The observed data are reported as estimated marginal means (SE) values, including the random effect and covariates within the model. * Difference (95% CI) and *p*-values are presented for the comparison between NHD and CHD, calculated with linear mixed model analysis of repeated measures. The overall 1-year effect is presented for the treatment interaction over time. Fixed factors included time, treatment and time * treatment interaction. Random intercepts included subject. In all models covariates gender, age and history of renal replacement therapy (RRT vintage) were added.

## Data Availability

Not applicable.

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
