# Peer review of "Nocturnal Hemodialysis Leads to Improvement in Physical Performance in Comparison with Conventional Hemodialysis"

_nutrients, 2022, doi:10.3390/nu15010168_

Round 1
Reviewer 1 Report
The present paper was of high interest. No information about the allocation is described: did you randomize them?
Did any patients start physical rehabilitation during this period? could you add this information?
Could you add information about the phlogosis marker or could you explain the reason because you did not consider them?
Why did you not add diastolic pressure, dialysis duration, and standardized kt/v week in the adjusted model? I agree with the clinical confounding added in the multivariate model, but statistical confounding variables should be chosen among the variables related to outcome and allocation and should be added. Did you perform correlation or regression tests at the baseline with allocation and outcomes?
Although the Linear mixed model was a valid statistical method, why did you not choose the GLM for repeated measures? Were there missing data?
Reviewer 2 Report
Thank you for the opportunity to review the manuscript.
I think the study is interesting and an important one to evaluate the physical performance and PEW among patients with CHD and NHD. Overall, a well study design with acknowledged of limitations, good statistical analysis tools used, good discussion, and proper acknowledgment of limitations.
Just an interesting fact that the mean age around 10 to 55 years which is a relatively young population for most patients with ESKD on dialysis.
Appreciate your study.
